# GenCO: Generating Diverse Solutions to Design Problems with Combinatorial Nature

## Abstract

Generating diverse objects (e.g., images) using generative models (such as GAN or VAE) has achieved impressive results in the recent years, to help solve many design problems that are traditionally done by humans. Going beyond image generation, we aim to find solutions to more general design problems, in which both the diversity of the design and conformity of *constraints* are important. Such a setting has applications in computer graphics, animation, industrial design, material science, etc, in which we may want the output of the generator to follow discrete/combinatorial constraints and penalize any deviation, which is non-trivial with existing generative models and optimization solvers. To address this, we propose GenCO, a novel framework that conducts end-to-end training of deep generative models integrated with embedded combinatorial solvers, aiming to uncover high-quality solutions aligned with nonlinear objectives. While structurally akin to conventional generative models, GenCO diverges in its role - it focuses on generating instances of combinatorial optimization problems rather than final objects (e.g., images). This shift allows finer control over the generated outputs, enabling assessments of their feasibility and introducing an additional combinatorial loss component. We demonstrate the effectiveness of our approach on a variety of generative tasks characterized by combinatorial intricacies, including game level generation and map creation for path planning, consistently demonstrating its capability to yield diverse, high-quality solutions that reliably adhere to user-specified combinatorial properties.

## 1 Introduction

Generating diverse and realistic objects with combinatorial properties is an important task with many applications. For example, in video game level design, we may want to generate a variety of levels that are both realistic and valid/playable (Zhang et al., 2020). Here, "valid" may refer to certain discrete characteristics of the level that must be satisfied (e.g., a minimum number of enemies, a path between the level entrance and exit, etc.). In automatic device design, we want to generate a variety of devices that meet foundry manufacturing constraints and optimize physics-related objectives (Schubert et al., 2022). In the design of new molecules, we want to generate a variety of chemically valid molecules with specific properties (Pereira et al., 2021). In all of these examples, the goal is to generate a variety of combinatorial solutions that are both feasible and high quality with respect to a given nonlinear objective. Another situation to consider is when the generated solutions do not have to strictly follow combinatorial constraints, but we want to discourage certain things as a penalty. For example, if we are creating various images of a map, we might want to ensure that the path planning in those maps is efficient (see section 4.2).

The difficulty in these settings arises from the fact that the generated objects need to satisfy specific discrete properties incorporated either as combinatorial constraints or penalties and be of high quality as evaluated by a nonlinear objective. The combinatorial constraints make it challenging to apply gradient-based methods to train standard generative models, such as GAN (Goodfellow et al., 2014), or VAE (Kingma & Welling, 2013), which may generate examples that violate the user-specified constraints. Additionally, the nonlinear objective makes it difficult to use traditional general-purpose combinatorial optimization tools since mixed-integer nonlinear programming (MINLP) is often slow in practice compared to the more heavily researched mixed integer linear programming (MILP), often requiring problem-specific solvers or leveraging specialized structure. Furthermore, traditional

optimization solvers are not geared towards generating diverse solutions and do not have the flexibility of deep learning models to quickly adapt to slightly modified settings. Recent work has attempted to address these issues by first training a generative model and then fixing/postprocessing the output after the fact (Zhang et al., 2020). However, these approaches are limited in that they only use the combinatorial solver to fix the output of the generative model and do not use the combinatorial solver in training. As a result, the generative model is not trained with the combinatorial solver in mind (i.e., end-to-end). Thus, the generator may fail to capture the distribution of high-quality feasible solutions as many of the generated examples may be "fixed" to a single solution. Additionally, some work constrains generative model outputs by penalizing the generator based on constraint violation (Chao et al., 2021). While this approach may work in some settings where the constraints must be satisfied, it is unclear how to extend this approach to more complex constraints, such as logical or general combinatorial constraints that can be readily expressed in MILP.

As a result, we propose GenCO that generates diverse solutions to design problems with combinatorial nature. GenCO combines the flexibility of deep generative models with the combinatorial efficiency of optimization solvers. Our approach mainly deals with two cases: (1) when the resulting solution needs to satisfy certain hard combinatorial constraints (**hard constraints**), and (2) the imposition of a desired penalty if the solutions violate the constraints (**soft constraints**). To achieve these two goals, we introduce a general concept known as the "combinatorial loss", which serves as an additional "restriction" on the generated solutions. In the context of combinatorial constraints, the combinatorial loss takes the form of a *projection* expressed as a mixed-integer linear program (MILP). In the second scenario, this general concept manifests as an extra penalty term within the objective function. This, coupled with the diversity mechanisms inherent in deep generative models, fine-tunes the generated solutions. This process both encourages diversity and ensures adherence to the specified combinatorial properties.

Our main contributions involve introducing a framework, GenCO, that serves two key purposes: 1) generating a range of solutions to nonlinear problems, ensuring they adhere to combinatorial properties; 2) integrating a combinatorial solver into the deep generative learning process. We showcase the effectiveness of our approach through a series of experiments on various generative combinatorial optimization problems. This includes tasks like game level design (section 4.1) and generating maps for path planning (section 4.2).

## 2 RELATED WORK

The interaction between generative models and combinatorial optimization has seen increased research interest as practitioners seek to integrate or draw inspiration from both paradigms. A prominent line of research has approached the problem of generating objects with combinatorial optimization in mind.

**Traditional constrained object generation**   Traditional constraint optimization methods were modified to search for multiple feasible solutions for problems concerning building layout (Bao et al., 2013), structural trusses (Hooshmand & Campbell, 2016), networks (Peng et al., 2016), building interiors (Wu et al., 2018), and urban design (Hua et al., 2019). Additionally, an approach based on Markov chain Monte Carlo (Yeh et al., 2012) samples objects that satisfy certain constraints. These approaches employ traditional sampling and optimization methods such as mathematical programming and problem-specific heuristics to obtain multiple feasible solutions. These methods often obtain multiple solutions by caching the feasible solutions found during the search process or modifying hyperparameters such as budgets or seed solutions. These methods can guarantee feasibility and optimality; however, they cannot synthesize insights from data such as historical good design examples or generate unstructured objects like images that are difficult to handle explicitly in optimization problems.

**Infeasibility penalization**   Further work has endeavored to modify deep generative models such as generative adversarial networks (Goodfellow et al., 2014) to penalize constraint violation. General purpose methods are proposed for generating objects respecting constraint graphs (Para et al., 2021) and blackbox constraints (Di Liello et al., 2020). Specialized approaches consider graph constraints where the generated object should be a graph that meets certain constraints that are specified by a blackbox function such as in the design of photonic crystals (Christensen et al., 2020), crystal

structure prediction (Kim et al., 2020), graph-constrained house generation (Tang et al., 2023), and general house plan generation (Nauata et al., 2021). In these settings, generating infeasible objects is penalized, and the generative model often has an inductive bias in the network architecture to generate feasible objects. These methods can use historical training datasets and handle unstructured objects; however, the output is often infeasible.

**Optimization-based priors** In a related direction, previous work proposed conditioning variational autoencoders (VAE) (Kingma & Welling, 2013) with combinatorial programs (Misino et al., 2022). Here, the latent information is extracted by the encoder and then fed through a logic program, made differentiable via DeepProblog (Manhaeve et al., 2018), such as performing handwritten operations on digits. The result is then fed through the decoder to generate the original image based on logical relationships. At test time, the goal is to generate objects that hopefully satisfy the logical relationship because the model was conditioned to do so. Here, the logical program helps to condition the generative model to generate objects based on patterns resulting from logical relationships and penalize cases where the logic is incorrect. Here, the generated objects are not guaranteed to satisfy the combinatorial constraints, but rather, the generative model has a logical structural prior. Previous work has made various optimization problems differentiable, such as quadratic programs (Amos & Kolter, 2017), probabilistic logic programs (Manhaeve et al., 2018), linear programs (Wilder et al., 2019a; Mandi & Guns, 2020; Elmachtoub & Grigas, 2017; Liu & Grigas, 2021), Stackelberg games (Perrault et al., 2020), normal form games (Ling et al., 2018), kmeans clustering (Wilder et al., 2019b), maximum likelihood computation (Niepert et al., 2021), graph matching (Rolínek et al., 2020), knapsack (Demirovic et al., 2019b;a), maxsat (Wang et al., 2019), mixed integer linear programs (Mandi et al., 2020; Paulus et al., 2021; Ferber et al., 2020), blackbox combinatorial solvers (Pogančić et al., 2020; Mandi et al., 2022; Berthet et al., 2020), nonlinear programs (Donti et al., 2017), continuous constraint satisfaction (Donti et al., 2020), cone programs (Agrawal et al., 2019). General-purpose methods are presented for minimizing downstream regret by learning surrogate loss functions (Shah et al., 2022; 2023; Zharmagambetov et al., 2023). Additionally, previous work has employed learnable linear solvers to solve nonlinear combinatorial problems (Ferber et al., 2023). Finally, a survey investigates the intersection between machine learning and optimization (Kotary et al., 2021). These approaches focus on identifying a single solution rather than generating diverse solutions. Additionally, many of these approaches are amenable for use by GenCO if a specific optimization problem better suits the generative problem.

**Enhacing combinatorial optimization with generative models** Recently, generative models have been proposed to improve combinatorial optimization. Zhang et al. (2022) use generative flow networks (gflownet) for robust scheduling problems. Sun & Yang (2023) use graph diffusion to solve combinatorial problems on graphs. Ozair et al. (2021) use vector quantized variational autoencoders to compress the latent space for solving planning problems in reinforcement learning. Zhao & You (2020) use generative adversarial networks to generate settings for sample average approximation in robust chance-constrained optimization. Additionally, in (Lopez-Piqueres et al., 2023), the authors generate continuous objects with tensor networks that satisfy linear constraints for optimization problems. These approaches target solving a fully specified optimization problem rather than generating objects with combinatorial constraints in mind.

## 3 GENCO: METHOD DESCRIPTION

### 3.1 MATHEMATICAL FORMULATION

The key distinction in our framework centers around the incorporation of a combinatorial loss, a departure from conventional deep generative models like Generative Adversarial Networks (GANs) illustrated in Figure 1. Rather than directly generating the ultimate object of interest, be it images or any other complex entity, our generator $G(\epsilon; \theta)$ parameterized by $\theta$ takes random noise $\epsilon$ and transforms it into a problem representation $c$ that encapsulates the underlying problem's essential features. This representation serves as a pivotal intermediary step. Subsequently, we feed $c$ to both the combinatorial loss and the generator loss (e.g. derived via a discriminator), ensuring that the solutions produced by the generator exhibit the desired characteristics: diversity, realism, and adherence to combinatorial properties.

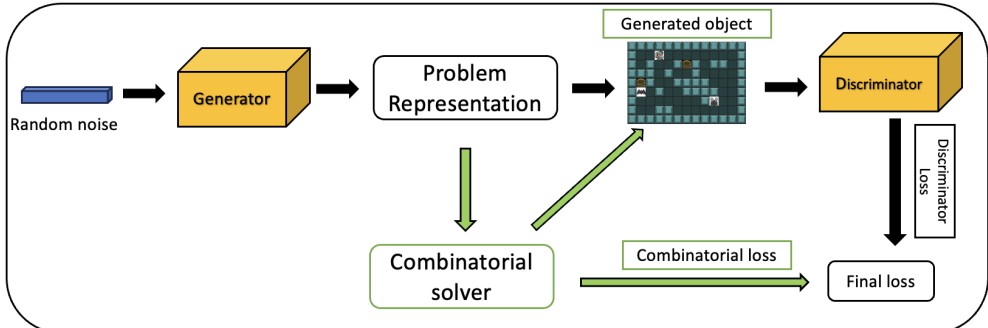

Figure 1: An overview of GenCO. The top part of the diagram corresponds to the ordinary "GAN-style" generative model (chosen as an example). By integrating the "combinatorial loss" into both the loss function and the object generation process (bottom part), we effectively encode the combinatorial nature of the problem.

To train such a generator, we define the following bi-objective function, effectively balancing the generative loss $\mathcal{L}_{\text{gen}}$ with the combinatorial loss $\mathcal{C}$:

$$\min_{\theta} \quad \mathbb{E}_{\epsilon}[\mathcal{L}_{\text{gen}}(c) + \gamma \mathcal{C}(h(c))]$$
$$\text{s.t.} \quad c = G(\epsilon; \theta), \tag{1}$$

where $\mathcal{L}_{\text{gen}}$ encapsulates the generative loss, responsible for promoting diversity and realism. Its form can vary substantially depending on the specific generative model employed. For example, a Wasserstein GAN might utilize the adversarial discrepancy between outputs, while a Variational Autoencoder (VAE) would make use of the Evidence Lower Bound (ELBO).

Conversely, the combinatorial loss $\mathcal{C}(\cdot)$, penalized by the weighting factor $\gamma$, is coupled with the downstream optimization solver $h(\cdot)$. This solver plays a crucial role in enforcing or encouraging combinatorial properties. For instance, one might seek to generate objects that must adhere to physical or logical constraints, all while minimizing a specified (linear) cost. In such a scenario, it becomes imperative to formulate an instance of a Mixed-Integer Linear Program (MILP) or another type of optimization problem with a well-defined feasible region. This problem instance then integrates seamlessly into the training pipeline, serving as the backbone of $\mathcal{C}$.

Below, we consider two potential and practical settings for $\mathcal{C}$, each tailored to address specific real-world applications.

### 3.2 CONSTRAINED GENERATOR

In this setting, we consider a problem domain where the goal is to train a generator that outputs *feasible* solutions, which natively handles diversity. To generate objects that are guaranteed to be feasible, we set $\gamma \to +\infty$ in equation 1 for any $c$ that does not satisfy constraints (defined by feasible region $\Omega$), 0 otherwise. To handle this directly with such a high value of $\gamma$ is computationally inefficient. Instead, we employ a "projection layer" denoted as $h(\cdot)$ that maps $c$ to its nearest feasible point in the feasible set $\Omega$. Specifically, we define $h(c) = \arg\max_{x \in \Omega} c^T x$ [1], where $c$ is an output of the generator $G(\epsilon; \theta)$. In this context, a generated problem description $c$ (as in Fig.1) can be viewed as an unconstrained generated object that is subsequently mapped via $h(\cdot)$ to $x$ to adhere to the defined constraints. With this approach, we can reformulate equation 1 into the following unconstrained optimization problem:

$$\min_{\theta} \quad \mathbb{E}_{\epsilon}[\mathcal{L}_{\text{gen}}(h(G(\epsilon; \theta)))]. \tag{2}$$

Note that the feasibility within region $\Omega$ is guaranteed to be satisfied due to the form of $h(\cdot)$. Directly minimizing this objective is possible thanks to the recent advancements in differentiable solvers literature (Agrawal et al., 2019; Pogančić et al., 2020; Sahoo et al., 2022).

---

[1] Although a dot product is a natural choice of proximity for cosine distance, one could employ other metrics, e.g. $|c - x|$

Algorithm 1 provides a high-level description of our method for this setting. It starts by initializing the generator parameters. In each iteration, a problem description is randomly selected using the generator. An optimization problem is then solved to find the solution that maximizes the similarity between $c$ within a defined feasible set ($x$ is the corrected/feasible version of $c$). Finally, the generator parameters are updated through backpropagation. We also provide an application of this generic algorithm to the concrete generative model examples of GANs and VQVAEs in Appendices B-C.

---

**Algorithm 1:** Generator Training in Constrained Setting

---
   **Output**: Trained generator
   Initialize generator parameters $\theta$;
   **while** not converged **do**
      Sample a noise $\epsilon$;
      Sample a problem description $c \sim G(\epsilon; \theta)$;
      Call a solver $x^* = \arg\max_{x \in \Omega} c^T x = h(c)$;
      Compute loss $= \mathcal{L}_{\text{gen}}(x^*)$;
      Backpropagate $\nabla_\theta$loss to update $\theta$;
   **end while**

---

## 3.3 Penalized Generator

The formulation from the previous section 3.2 naturally extends to penalty form, i.e., we *encourage* generated problem representation $c$ to respect certain combinatorial properties. In this scenario, we can directly leverage equation 1 as there is no need in projection layer:

$$\min_\theta \quad \mathbb{E}_\epsilon [\mathcal{L}_{\text{gen}}(G(\epsilon; \theta)) + \gamma \mathcal{C}(h(G(\epsilon; \theta)))]. \tag{3}$$

Here, we seamlessly integrate the generator into the objective function. While it is true that we do not impose any explicit constraints on the output of the generator, and one could use the output of $G$ directly, we contend that introducing a *combinatorial penalty or loss* as a penalty remains a crucial framework with significant practical utility. This approach serves to further guide the generation process and injects domain-specific knowledge that is easy to interpret, ensuring that the generated solutions exhibit desired combinatorial properties, a facet that holds immense practical relevance.

For instance, we may aim to generate images tailored for path planning in strategy games. These maps feature various types of terrains, each incurring distinct costs when placed on the map. For efficient path planning, our goal is to find a "low cost" *shortest path* between two endpoints. Therefore, in this scenario, $\mathcal{C}$ incorporates a domain specific (fixed) mapping (e.g. a neural net) that transforms an image into a graph with corresponding weights assigned to each edge, followed by solving the shortest path problem (e.g. using Dijkstra's algorithm).

However, in this particular case, there is no need to introduce a "projection" operator as in the previous case since there are no explicit constraints that must be strictly satisfied. We can directly employ gradient-based optimization, allowing for differentiation through the combinatorial problem. This is achieved approximately by employing methods outlined in (Pogančić et al., 2020; Sahoo et al., 2022). Pseudocode 2 describes the application of GenCO for penalty setting. Furthermore, Appendix A.1 provides an application of this generic algorithm to a concrete example of GANs.

It is crucial to emphasize that the given penalized setting deviates from the conventional decision-focused learning (DFL) or smart predict-then-optimize paradigms (Elmachtoub & Grigas, 2017; Donti et al., 2017). In our approach, we do not possess access to the ground truth costs $c$, and the ultimate loss function encompasses both the generator's loss and the output of a solver.

## 4 Experiments

We tested our approach on two applications: level generation for the Zelda game and map generation for path planning in Warcraft. Although both applications are in the game domain, they have quite different use cases. Nevertheless, both settings involve combinatorial optimization in the pipeline, which makes the typical use of deep generative models nontrivial. In these experiments,

---

**Algorithm 2:** Generator Training in Penalty Setting

---

   **Output**: Trained generator
   Initialize generator parameters $\theta$;
   (optional) Define a fixed mapping $g(\cdot)$ for $c$
   **while** not converged **do**
      Sample a noise $\epsilon$;
      Sample a problem description $c \sim G(\epsilon; \theta)$;
      (optional) apply a mapping $g(\cdot)$ to transform $c$;
      Call a solver $x^* = h(g(c))$ ;
      Compute loss $= \mathcal{L}_{\text{gen}}(c) + \gamma \mathcal{C}(x^*)$;
      Backpropagate $\nabla_\theta$loss to update $\theta$;
   **end while**

---

we demonstrated that GenCO *significantly outperforms the baselines, efficiently finding diverse, realistic solutions that obey combinatorial properties*. This success paves the way for combining combinatorial optimization with deep generative models.

## 4.1 GENERATING GAME LEVELS USING GANS

We evaluate our model on the task of generating diverse Zelda game levels. In this setting, we use GenCO to train using a GAN-like *constrained* formulation (refer to section 3.2 and Appendix B).

### 4.1.1 SETTINGS

**Explicitly Constrained GAN: Game Level Design**   We train GenCO on the task of generating Zelda game levels. Here, we are given examples of human-crafted game levels and are asked to generate fun new levels. The generated levels must be playable in that the player must be able to solve them by moving the character through a route that reaches the destination. Additionally, the levels should be realistic in that they should be somewhat similar to the real game levels as measured by an adversary that is trained to distinguish between real and fake images ($\mathcal{L}_{\text{gen}}$). We use the same dataset as (Zhang et al., 2020), consisting of 50 Zelda game levels, as well as the same network architectures, as we don't tune the hyperparameters for our model in particular but rather compare the different approaches, all else equal.

Here, we evaluate several approaches, including the previous work (Zhang et al., 2020), which we call *GAN + MILP fix*. This approach first trains a standard Wasserstein GAN architecture to generate the game levels. Specifically, they train the WGAN by alternatively training two components: a generator and a discriminator. The generator is updated to "fool" the discriminator as much as possible in that it tries to maximize the loss of the discriminator. The discriminator tries to correctly separate the generated and real game levels into their respective classes. When the practitioner then wants to generate a valid level, this approach generates a level with the generator and then fixes it using a MILP formulation that finds the nearest feasible game level where proximity is determined by cosine distance, which is equivalent to minimizing a dot product distance.

We evaluate two variants of GenCO, *GenCO - Fixed Adversary*, which iteratively updates the generative model to fool a fixed pretrained adversary, and *GenCO - Updated Adversary*, which updates both the generator and adversary during training. The fixed adversary approach trains to fool a pretrained adversary that is obtained from the fully trained GAN from previous work (Zhang et al., 2020). Both GenCO approaches are initialized with the fully trained GAN from previous work (Zhang et al., 2020).

### 4.1.2 RESULTS

We present a table of results in Table 1 and examples in Figure 2, which includes the performance of the previous approach as well as two variants of our proposed approach. In these settings, we estimate performance based on sampling 1000 levels. Each level is made out of a grid, with each grid cell having one of 8 components: wall, empty, key, exit, 3 enemy types, and the player. A valid level is one that can be solved by the player in that there is a valid route starting at the player's location,

| Approach | % unique | GAN loss ($\mathcal{L}_{gen}$) (lower better) | GenCO adversary (lower better) |
|---|---|---|---|
| GenCO - Updated Adversary | 0.995 | -10.10 | -4.49 |
| GenCO - Fixed Adversary | 0.22 | -1.45 | -0.85 |
| GAN + MILP fix (Previous Work) | 0.52 | 0.22 | 0.24 |

Table 1: Game level design comparison. We compare GenCO with an updating adversary against GenCO with a fixed adversary, and lastly, previous work that postprocesses solutions to be valid game levels. Here, all levels are valid, but GenCO exhibits more diversity and realism.

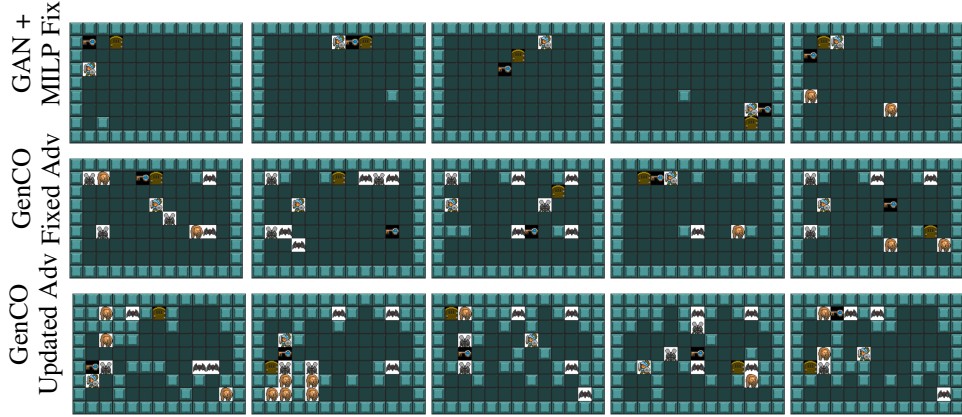

Figure 2: Generated zelda game examples. The GenCO Updated Adversary obtains solutions that seem more realistic than mainly empty fixed GAN instances and fixed adversary instances.

collecting the key, and then reaching the exit. We evaluate the performance of the models using two types of metrics: diversity, as measured by the percentage of unique levels generated, and fidelity, as measured by the average objective quality of a fixed GAN adversary. We evaluate using adversaries from both the previous work and GenCO. As shown in Table 1, we find that GenCO with an updating adversary generates unique solutions at a much higher rate than previous approaches and also generates solutions that are of higher quality as measured by both the GAN adversary and its own adversary. The adversary quality demonstrates that the solutions are realistic in that neither the adversary from the previous work nor from GenCO is able to distinguish the generated examples from the real examples. This is further demonstrated in Figure 2 with the updated adversary generating realistic and nontrivial game levels. Furthermore, given that the levels are trained on only 50 examples, we can obtain many more game levels. Note here that both approaches are guaranteed to give playable levels as they are postprocessed to be valid. However, GenCO is able to generate more diverse solutions that are also of higher quality.

**Uniqueness** As shown in Table 1, GenCO, with an updated adversary, obtains the highest percentage of unique solutions, generating 995 unique solutions out of 1000. This is significantly higher than the previous work, which only generated 520 unique solutions. This is likely due to the fact that the generator is trained with the downstream fixing explicitly in the loop. This means that while the previous work may have been able to "hide" from the adversary by generating slightly different continuous solutions, these continuous solutions may project to the same discrete solution. On the other hand, by integrating the fixing into the training loop, GenCO's generator is unable to hide in the continuous solution space and thus is heavily penalized by generating the same solution as the adversary will easily detect those to be originating from the generator. In essence, this makes the adversary's task easier as it only needs to consider distinguishing between valid discrete levels rather than continuous and unconstrained levels. This is also reflected in the adversary quality, where GenCO's adversary is able to distinguish between levels coming from the previous work's generator and the real levels with a much better loss.

### 4.2 MAP GENERATION FOR PATH PLANNING

In this experiment, we consider somewhat different and more challenging for GenCO setting of generation of image maps for strategy games like Warcraft. This poses a greater challenge because the generator $G$ described in equation 1 is not restricted and can produce any image of a map (setting descrbied in section 3.3). However, we are required to encourage $G$ to generate a map with two main criteria: 1) it should resemble real and diverse game maps, as determined by an adversary trained to differentiate between real and fake images; 2) *the cost of the shortest path from the top-left to the bottom-right corners (source and destination) should be minimized*. Intuitively, it corresponds to (mostly) a diagonal part of the image, and a map can include various elements (terrains) like mountains, lakes, forests, and land, each with a specific cost (for example, mountains may have a cost of 3, while land has a cost of 0). To calculate this, we pass the generated image to the fixed ResNet (called "cost NN") to get the graph representation together with edge costs. The cost NN corresponds to the mapping $g(\cdot)$ in Algorithm 2. This is then followed by the shortest path solver ($h(\cdot)$ in Algorithm 2). The objective is to populate the map with a diverse range of objects while ensuring the shortest path remains low-cost. We use the same dataset as in (Pogančić et al., 2020) with DCGAN architecture adapted from (Zhang et al., 2020). More implementation details and experimental settings can be found in Appendix A.

#### 4.2.1 BASELINES

We examine various baseline models, including an "Ordinary GAN" that does not incorporate the Shortest Path objective (see the top part of Fig. 1). Like the Zelda experiment, this approach employs a standard Wasserstein GAN architecture closely following (Zhang et al., 2020). However, in this case, we generate images directly instead of encoding game levels. More precisely, we train the WGAN by iteratively training two components: a generator and a discriminator. The generator is fine-tuned to deceive the discriminator to the greatest extent possible, aiming to maximize the discriminator's loss. The discriminator, on the other hand, endeavors to accurately distinguish between generated and authentic game levels and categorize them accordingly.

The next baseline is "GAN + cost NN", which simultaneously feeds the generator's output into both the discriminator and a ResNet that calculates the *costs* for each edge in the grid. We then average the output to obtain the final loss for generator. We then average these outputs to derive the ultimate loss for the generator. While this approach considers costs associated with objects, it does not incorporate information about the Shortest Path.

As for the GenCO, it generalizes both of these approaches incorporating combinatorial solver into the pipeline. The detailed algorithm is given in Appendix A. To backpropagate through the solver, we employ the "identity with projection" method from (Sahoo et al., 2022). The remaining settings are similar to "Ordinary GAN".

#### 4.2.2 RESULTS

Quantitative results are showcased in Table 2. The "Ordinary GAN" focuses exclusively on the GAN's objective, without considering the Shortest Path's objective $f$. In contrast, GenCO achieves higher performance with regards to the SP's objective, albeit with a slight reduction in GAN's loss. On the other hand, the "GAN + cost NN" approach optimizes using the cost vector of the entire map (generated by ResNet). In other words, it tries to uniformly avoid placing costly objects in any part of the image, whereas we are interested only in the shortest path. While it demonstrates a modest enhancement in the objective $f$, it experiences a notable decline in terms of generator loss ($\mathcal{L}_{gen}$). This indicates a trade-off between optimizing for the Shortest Path and the GAN's objective.

Such quantitative results directly translate into image qualities. In fig. 3, a subset of generated Warcraft map images using various methods is displayed. The "Ordinary GAN" tends to generate maps with elements such as mountains and lakes, which are considered "very costly", particularly along the Shortest Path from the top-left to the bottom-right (which mostly goes through the diagonal). This makes sense since WGAN is trained with no information about grid costs. In contrast, the "GAN + cost NN" approach produces less costly maps, albeit with reduced diversity. For instance, most part of the image is populated with the same object. On the other hand, GenCO strikes a balance, achieving a cheap Shortest Path while maintaining a diverse range of elements on the map. The Shortest Path is efficient, and the map exhibits a rich variety of features simultaneously.

| Approach | GAN loss ($\mathcal{L}_{gen}$) (lower better) | SP loss ($\mathcal{C}$) (lower better) |
|---|---|---|
| Ordinary GAN | **0.6147** | 36.45 |
| GAN + cost NN | 0.8994 | 35.61 |
| **GenCO (ours)** | 0.6360 | **23.99** |

Table 2: Performance comparison on Warcraft map generation. Our here goal is to create a map that is both realistic ("fools" discriminator, i.e. lower GAN loss) and has small objective for the shortest path (SP). Results are averaged across 100 instances.

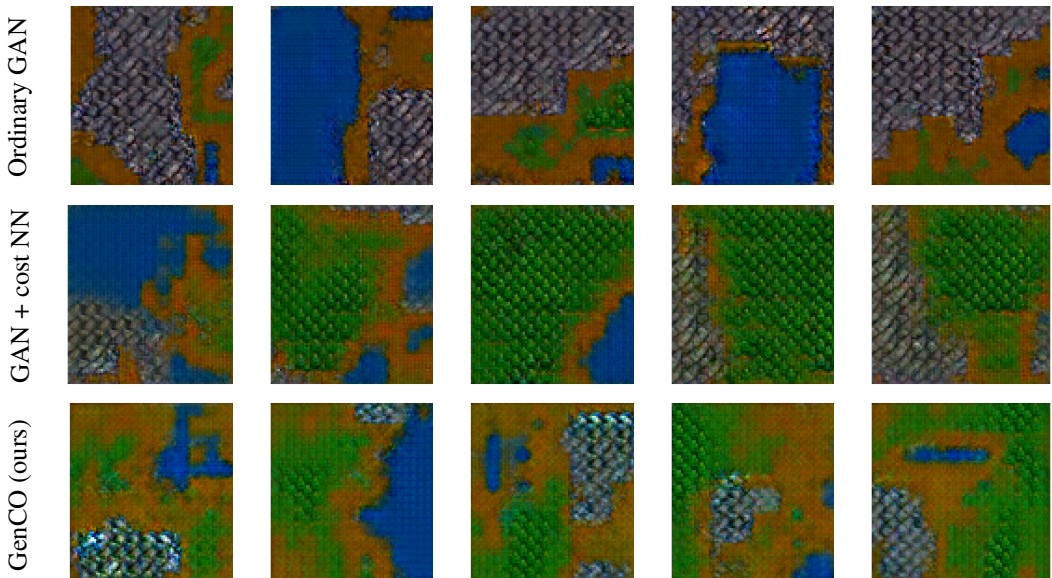

Figure 3: A subset of generated Warcraft map images using various methods. "Ordinary GAN" generates "very costly" maps (e.g. mountains, lakes) along the Shortest Path from top-left to bottom-right; "GAN + cost NN" generates less costly maps but they are *less diverse*; GenCO: SP path is cheap and the map is diverse at the same time.

## 5 CONCLUSION

In this paper, we introduce GenCO, a framework for integrating combinatorial constraints in a variety of generative models, and show how it can be used to generate diverse combinatorial solutions to nonlinear optimization problems. Unlike existing generative models and optimization solvers, GenCO guarantees that the generated diverse solutions satisfy combinatorial constraints, and we show empirically that it can optimize nonlinear objectives.

The underlying idea of our framework is to combine the flexibility of deep generative models with the guarantees of optimization solvers. By training the generator end-to-end with a surrogate linear combinatorial solver, GenCO generates diverse and combinatorially feasible solutions, with the generative loss being computed only on feasible solutions.

We have tested GenCO on various combinatorial optimization problems and generative settings, including GAN in Zelda game level generation and Warcraft map generation for path planning, demonstrating the flexibility of our framework for integrating into different generative paradigms. Our framework consistently produced diverse and high-quality solutions that satisfy the combinatorial constraints, which can be flexibly encoded using general-purpose combinatorial solvers.

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

# A  DETAILED EXPERIMENTAL SETUP FOR MAP GENERATION

## A.1  PSEUDOCODE

---
**Algorithm 3:** GenCO Penalized generator training for GANs.

---
1: Initialize generator parameters $\theta_{gen}$;
2: Initialize adversary parameters $\theta_{adv}$;
3: Input: distribution of the true dataset ($c_{true} \sim p_{\text{data}(c)}$), GAN's objective $\mathcal{L}(\theta_{gen}, \theta_{adv})$;
4: **for** epoch $e$ **do**
5:     **for** batch $b$ **do**
6:         Sample a noise $\epsilon$;
7:         Sample true examples from dataset $c_{true} \sim p_{\text{data}}(c)$;
8:         Sample fake examples using $c_{fake} \sim G(\epsilon; \theta_{gen})$;
9:         Transform $c_{fake}$ into the coefficients of the optimization problem: $c = g(c_{fake})$;
10:       Solve: $x^* = \arg\min_{x \in \Omega} c^T z$;
11:       Backpropagate $\nabla_{\theta_{gen}} \left[ \mathcal{L}(c_{fake}; \theta_{gen}, \theta_{adv}) + \beta c^T x^* \right]$ to update $\theta_{gen}$;
12:       Backpropagate $\nabla_{\theta_{adv}} \left[ -\mathcal{L}(c_{fake}; \theta_{gen}, \theta_{adv}) + \mathcal{L}(c_{real}; \theta_{gen}, \theta_{adv}) \right]$ to update $\theta_{adv}$;
13:     **end for**
14: **end for**

---

Algorithm 3 provides a detailed description of the GenCO framework in our penalty formulation and utilized in section 4.2. In this process, we sample both real and synthetic data, drawing from the true data distribution and the generator $G$ respectively (lines 7–8). Subsequently, the synthetic data undergoes a fixed mapping (e.g. ResNet in our experiments), called cost neural net (or cost NN), to obtain coefficients for the optimization problem, specifically the edge weights for the Shortest Path. Following this, we invoke a solver that provides us with a solution and its associated objective (lines 10–11). We then proceed to update the parameters of the generator $G$ using both the GAN's objective and the solver's objective. Finally, we refine the parameters of the adversary (discriminator) in accordance with the standard GAN's objective.

## A.2  SETTINGS

We employ ResNet as the mapping $g(\cdot)$ from Algorithm 3, which transforms an image of a map into a $12 \times 12$ grid representation of a weighted directed graph: $g : \Re^{96 \times 96 \times 3} \to \Re^{12 \times 12}$. The first five layers of ResNet18 are pre-trained (75 epochs, Adam optimizer with lr=$5e - 4$) using the dataset from Pogančić et al. (2020), comprising 10,000 labeled pairs of image–grid (refer to the dataset description below). Following pretraining, we feed the output into the Shortest Path solver, using the top-left point as the source and the bottom-right point as the destination. The resulting objective value from the Shortest Path corresponds to $f$.

**Dataset**  The dataset used for training in the Shortest Path problem with $k = 12$ comprises 10,000 randomly generated terrain maps from the Warcraft II tileset Pogančić et al. (2020) (adapted from Guyomarch (2017)). These maps are represented on a $12 \times 12$ grid, with each vertex denoting a terrain type and its associated fixed cost. For example, a mountain terrain may have a cost of 9, while a forest is assigned a cost of 1. It's important to note that in the execution of Algorithm 3, we don't directly utilize the actual (ground truth) costs, but rather rely on ResNet to generate them.

## A.3  ARCHITECTURE

We employ similar DCGAN architecture taken from Zhang et al. (2020) (see fig. 3 therein). Input to generator is 128 dimensional vector sampled from Gaussian noise centered around 0 and with a std of 1. Generator consists of five (256–128–64–32–16) blocks of transposed convolutional layers, each with $3 \times 3$ kernel sizes and batch normalization layers in between. Discriminator follows by mirroring the same architecture in reverse fashion. The discriminator mirrors this architecture in reverse order. The entire structure is trained using the WGAN algorithm, as described inZhang et al. (2020).

## B  GANS WITH COMBINATORIAL CONSTRAINTS

In the generative adversarial networks (GAN) setting, the generative objectives are measured by the quality of a worst-case adversary, which is trained to distinguish between the generator's output and the true data distribution. Here, we use the combinatorial solver to ensure that the generator's output is always feasible and that the adversary's loss is evaluated using only feasible solutions. This not only ensures that the pipeline is more aligned with the real-world deployment but also that the discriminator doesn't have to dedicate model capacity to detecting infeasibility as indicating a solution is fake and instead dedicate model capacity to distinguishing between real and fake inputs, assuming they are all valid. Furthermore, we can ensure that the objective function is optimized by penalizing the generator based on the generated solutions' objective values:

$$\mathcal{L}_{\text{gen}}(G(\epsilon; \theta_{gen})) = \mathbb{E}_{\epsilon}\left[\log(1 - f_{\theta_{adv}}(G(\epsilon; \theta_{gen})))\right] \tag{4}$$

where $f_{\theta_{adv}}$ is an adversary (a.k.a. discriminator) and putting this in the context of equation 2 leads to:

$$\min_{\theta_{gen}} \quad \mathcal{L}_{\text{gen}}(S(\tilde{G}(\epsilon; \theta_{gen}))) = \mathbb{E}_{\epsilon}\left[\log(1 - f_{\theta_{adv}}(S(\tilde{G}(\epsilon; \theta_{gen}))))\right] \tag{5}$$

where $\tilde{G}$ is unconstrained generator and $S$ is a surrogate combinatorial solver as described above. Here, we also have adversary's learnable parameters $\theta_{adv}$. However, that part does not depend on combinatorial solver and can be trained as in usual GAN's. The algorithm is presented in pseudocode 4.

---

**Algorithm 4:** GenCO in the constrained generator setting

---

Initialize generator parameters $\theta_{gen}$;
Initialize adversary parameters $\theta_{adv}$;
**for** epoch $e$ **do**
    **for** batch $b$ **do**
        Sample problem $\epsilon$;
        Sample true examples from dataset $x_{true} \sim p_{\text{data}}(x)$;
        Sample linear coefficients $c \sim G(\epsilon; \theta_{gen})$;
        Solve $x^* = \arg\max_{x \in \Omega} c^T x$;
        Backpropagate $\nabla_{\theta_{gen}} \mathcal{L}_{\text{gen}}(x^*; \theta_{gen})$ to update generator (equation 5);
        Backpropagate $\nabla_{\theta_{adv}} \left[\log(f_{\theta_{adv}}(x_{true})) - \mathcal{L}_{\text{gen}}(x^*; \theta_{gen})\right]$ to update adversary;
    **end for**
**end for**

---

## C  GENCO – VQVAE

The formulation below spells out the VQVAE training procedure. Here, we simply train VQVAE on a dataset of known objective coefficients, which solves the problem at hand. A variant of this also puts the decision-focused loss on the generated objective coefficients, running optimizer $g_{\text{solver}}$ on the objective coefficients to get a solution and then computing the objective value of the solution.

$$L_{\text{ELBO}}(c, \theta, E) = \mathbb{E}_{q_{\theta}(z|c)}[\log p_{\theta}(c|z)] - \beta \cdot D_{\text{KL}}(q_{\theta}(z|c)||p(z)) + \gamma \cdot \|sg(\mathbf{e}_k) - \mathbf{z}_{e,\theta}\|_2^2 \tag{6}$$

Here $z$ is an embedding vector, $c$ is the objective coefficients, $\log p_{\theta}(c|z)$ is a loss calculated via the mean squared error between the decoder output and the original input objective coefficients, $q_{\theta}(z|c)$ is the encoder, $p(z)$ is the prior, and $sg(\cdot)$ is the stop gradient operator, $E$ is a discrete codebook that is used to quantize the embedding.

$$\mathcal{L}_{\text{optimization}} = \mathbb{E}_{c \sim p_{\theta}(c|z)}\left[f_{\text{obj}}(g_{\text{solver}}(c; y))\right] \tag{7}$$

The algorithm below maximizes a combination of the losses in Equation equation 6 and Equation equation 7.

---

**Algorithm 5:** Constrained generator Training for VQVAE

---

**Input**:Training data distribution $D$ over problem info $y$ and known high-quality solutions $x$,
  regularization weight $\beta$, linear surrogate solver $g_{\text{solver}}$, nonlinear objective $f_{\text{objective}}$.

**Output**:Trained encoder $f_{\text{enc}}$, decoder $f_{\text{dec}}$, and codebook $E$

Initialize the parameters of the encoder $f_{\text{enc}}$, decoder $f_{\text{dec}}$, and the codebook
  $E = \{e_1, e_2, \ldots, e_K\}$ with $K$ embedding vectors;
**for** $t = 1$ **to** $T$ **do**
 | Sample $y, x$ from the distribution $D$;
 | Compute the encoder output $z_e = f_{\text{enc}}(y, x)$;
 | Find the nearest embedding vector $z_q = \arg\min_{e \in E} \|z_e - e\|_2^2$;
 | Compute the quantization loss $L_{\text{quant}} = \|z_e - z_q\|_2^2$;
 | Decode the embedding $\tilde{c} = f_{\text{dec}}(y, z_q)$;
 | Solve $\tilde{x} = \arg\max_{x \in \Omega} c^T x$;
 | Compute the reconstruction loss $L_{\text{recon}} = \|x - \tilde{x}\|_2^2$;
 | GenCo Variant: Compute the optimization loss $L_{\text{opt}} = f_{\text{objective}}(\tilde{x})$;
 | Compute the total loss: $L_{\text{total}} = L_{\text{recon}} + \beta_1 L_{\text{quant}} + \beta_2 L_{\text{opt}}$;
 | Update the parameters of the encoder, decoder, and codebook to minimize $L_{\text{total}}$;
**end**

---

## D   RESULTS FOR WARCRAFT WITH DENSITY/COVERAGE METRICS

| Approach | density (higher better) | coverage (higher better) | SP loss ($\mathcal{C}$) (lower better) |
|---|---|---|---|
| Ordinary GAN | 0.81 | 0.98 | 36.45 |
| GAN + cost NN | 1.09 | 0.98 | 35.61 |
| **GenCO (ours)** | 0.94 | 0.93 | **23.99** |

Table 3: Performance comparison on Warcraft map generation. Our goal here is to create a map that is both realistic/diverse (has higher density/coverage metrics) and has a small objective for the shortest path (SP). Metrics are computed based on 100 random instances.

## E   DIVERSE INVERSE PHOTONIC DESIGN

To demonstrate that our approach is more broadly effective, we demonstrate improvement over a generative + postprocess baseline for an inverse photonic device design setting.

The inverse photonic design problem Schubert et al. (2022) asks how to design a device consisting of fixed and void space to route wavelengths of light from an incoming location to desired output locations at high intensity. Here the feasible region consists of satisfying manufacturing constraints that a die with a specific shape must be able to fit in every fixed and void shape. Specifically, the fixed and void regions respectively should able to be represented as a union of the die shape. The objective function here consists of a nonlinear but differentiable simulation of the light using Maxwell's equations. Previous work demonstrated an approach for finding a single optimal solution to the problem. However, we propose generating a diverse collection of high-quality solutions using a dataset of known solutions.

Here, we instantiate GenCO using a vector quantized variational autoencoder (VQVAE) Van Den Oord et al. (2017) generative backbone. Here the autoencoder is fed in a known solution then uses neural networks for the encoder and decoder. The continuous decoded object is then fed into the constrained optimization layer to enforce that the generated solution is feasible. This feasible solution is then used to calculate the reconstruction loss. Furthermore, in this setting we have a penalty term that consists of the simulation of Maxwell's equations. As such, we ablate the data distribution approximation, and penalty components of GenCO: whether or not to train using the reconstruction loss, and whether or not to penalize generated solutions based on Maxwell's equations. We consider

| Approach | % Unique (higher better) | Avg Solution Loss (lower better) | Density (higher better) | Coverage (higher better) |
|---|---|---|---|---|
| VQVAE + postprocess | 30.6% | 1.244 | 0.009 | 0.006 |
| GenCO (reconstruction only) | 100% | 1.155 | 0.148 | 0.693 |
| GenCO (objective only) | 46.6% | 0 | 0.013 | 0.036 |
| GenCO (reconstruction + objective) | 100% | 0 | 0.153 | 0.738 |

Table 4: Comparison table for inverse photonic design evaluating variants of GenCO with a VQ-VAE generative backbone against the same model architecture which postprocesses solutions to be feasible. Solution loss is evaluated using a simulation of Maxwell's equations as in previous work.

a baseline here of training the same generative architecture without the combinatorial optimization layer and then postprocessing generated examples during evaluation using a combinatorial solver. We demonstrate results below including the percentage of unique discrete solutions that are generated, the average loss evaluated using Maxwell's equations, as well as the density and coverage with respect to the training dataset. The dataset of 100 examples is obtained by expensively running previous work [1] until it reaches an optimal loss 0 solution. We evaluate performance on generating 1000 feasible examples.

These results in Table 4 demonstrate that the postprocessing approach obtains very few unique solutions which all have high loss and furthermore don't cover the data distribution well. This is largely due to the method not being trained with the postprocessing end-to-end. Although it closely approximates the data distribution with continuous and infeasible objects, when these continuous objects are postprocessed to be made feasible, they are no longer representative of the data distribution and many continuous solutions collapse to the same discrete solution.

Here GenCO - reconstruction only gives many unique solutions that closely resemble the data distribution. However, the generated devices fail to perform optimally in the photonic task at hand. Disregarding the reconstruction loss and only training the decoder to generate high-quality solutions yields high-quality solutions but which are not diverse. Combining both the generative reconstruction penalty as well as the nonlinear objective, GenCO is able to generate a variety of unique solutions that optimally solve the inverse photonic design problem and have good density and coverage for the data distribution.

