# OpenReview forum: "GenCO: Generating Diverse Solutions to Design Problems with Combinatorial Nature"
_ICLR.cc/2024/Conference — Submitted to ICLR 2024_

### Official Review · Reviewer_GDcb · 2023-10-29

**Soundness:** 2 fair
**Presentation:** 2 fair
**Contribution:** 2 fair
**Rating:** 5
**Confidence:** 3

**Summary:**

GenCO" is a novel framework that integrates deep generative models with combinatorial solvers to address design challenges that require diverse solutions while adhering to specific constraints. Unlike conventional generative models, GenCO focuses on generating instances of combinatorial optimization problems, allowing finer control over the generated outputs and introducing an additional combinatorial loss component. The framework's effectiveness is demonstrated across various generative tasks, consistently producing diverse, high-quality solutions that meet user-defined combinatorial properties.

**Strengths:**

Combing the algorithmic prior of a combinatorial optimization solver with various generative models is an effective and promising approach.

**Weaknesses:**

The application methodology is relatively straightforward, involving the imposition of constraints or the assignment of optimization performance metrics to existing proposed methods, which are well-established techniques.

**Questions:**

1. How does this research address the mechanism behind generating diverse solutions when assigning combinatorial properties to various generative models, and does it simply rely on the use of generative models to claim diversity?

2. Combining GAN and VAE within a unified framework may seem unusual, given their distinct purposes. Is there a thoughtful consideration of the different objectives of these two generative models, or does it feel mechanically extended without such reflection?

3. Could you clearly define L_{Gen} for both VAE and GAN and provide separate descriptions for each in Figure 1? If the proposed learning method is more suitable for one of the generative models, focusing on it may be beneficial.

4. In section 3.2, regarding the Constrained Generator, are there any limitations in terms of the expressive power of linear projection? How does it differ from techniques like projected gradient ascent commonly used in constrained optimization?

5. In the penalized generator discussed in section 3.3, the idea of directly using the cost obtained through the combinatorial optimization solver h for model training has been present in prior research. Are there any novel aspects from a learning perspective in this new approach?

---

> ### Author Response · Authors · 2023-11-18
>
> > The application methodology is relatively straightforward, involving the imposition of constraints or the assignment of optimization performance metrics to existing proposed methods, which are well-established techniques.
>
> Please refer to our general response (1), where we clarify the novelty of our approach and main contributions.
>
>
> > How does this research address the mechanism behind generating diverse solutions when assigning combinatorial properties to various generative models, and does it simply rely on the use of generative models to claim diversity?
>
> Our research indeed leverages generative models to achieve diversity in solutions by approximating the underlying data distribution. The mechanism involves training the generative model to capture and represent the diverse characteristics inherent in the given combinatorial problem.
>
>
> > Combining GAN and VAE within a unified framework may seem unusual, given their distinct purposes. Is there a thoughtful consideration of the different objectives of these two generative models, or does it feel mechanically extended without such reflection?
>
> Thank you for raising this important point. While GANs and VAEs traditionally serve distinct purposes, our approach aims to highlight the flexibility of combinatorial optimizers in accommodating both GAN and VAE frameworks. The integration is not merely a mechanical extension; rather, it's a thoughtful exploration of the adaptability of combinatorial optimizers to different generative models. We acknowledge the distinct objectives of GANs and VAEs and demonstrate that our approach GenCO is versatile enough to be applicable to both, as demonstrated in Appendix C. Furthermore, we evaluate a new setting of generating inverse photonic designs in which we employ a VQVAE as the generative backbone and ensure that decoded objects satisfy the combinatorial constraints during training.
>
>
> > Could you clearly define L_{Gen} for both VAE and GAN and provide separate descriptions for each in Figure 1? If the proposed learning method is more suitable for one of the generative models, focusing on it may be beneficial.
>
> For VAE, the L_{Gen} is equivalent to the ELBO loss that VAE is ultimately optimizing. For GAN, the loss would be equivalent to the KL divergence or Wasserstein loss that is being optimized. These losses are implicitly defined by the different methods by generally comparing the generated object to the data distribution in some way, reconstruction error for the VAE, and discrimination error for the GANs. Ultimately, this is a loss imposed on the distribution represented by the generative model, informing how aligned the generative model is with the data distribution. Please refer to Appendix C for the complete description of our approach for VAEs.
>
>
> > In section 3.2, regarding the Constrained Generator, are there any limitations in terms of the expressive power of linear projection? How does it differ from techniques like projected gradient ascent commonly used in constrained optimization?
>
> We would like to note that it is a projection in the sense that we are identifying a combinatorially constrained discrete solution that is near the original continuous solution (i.e. projection into combinatorial space). Because our settings have only binary decisions, we can guarantee that each solution is reachable by some continuous solution (meaning that all discrete solutions are possible to generate). However, we would note that the projection is not linear (i.e., not a simple linear transformation). On the contrary, the projection is the solution to a discrete linear optimization problem, which is highly discrete and piecewise constant in that a large area in the original continuous space may map to the same solution in discrete space. Additionally, while projected gradient ascent is a popular approach for solving some constrained optimization problems, it is not generally used in combinatorial optimization where the standard methods would be a variant of tree search for MILP and constraint satisfaction problems or variants of simplex for linear programs.
>
>
>
> > In the penalized generator discussed in section 3.3, the idea of directly using the cost obtained through the combinatorial optimization solver h for model training has been present in prior research. Are there any novel aspects from a learning perspective in this new approach?
>
> While we acknowledge prior works that incorporate costs as penalties, such as Donti et al. 2020, it's important to highlight the novelty of our approach from a learning perspective. Existing works typically focus on the continuous case or rely on continuous relaxation, whereas our method directly handles combinatorial penalties or losses without such relaxation and trains the whole pipeline end-to-end. This distinction represents a novel aspect of our approach, as we are not aware of any previous work that addresses combinatorial penalties in the same direct manner.

---

> > ### Comment · Reviewer_GDcb · 2023-11-23
> > **Thank you for your clarification.**
> >
> > Many of my concerns have been addressed. I raised my score to 5.

---

### Official Review · Reviewer_X2M2 · 2023-10-31

**Soundness:** 2 fair
**Presentation:** 2 fair
**Contribution:** 2 fair
**Rating:** 6
**Confidence:** 3

**Summary:**

The paper introduces GenCO, a framework to generate diverse and high-quality solutions that satisfy combinatorial constraints which is an important factor in design problems where both diversity of solutions and adherence to constraints are important. GenCO combines  the flexibility of deep generative models with the combinatorial efficiency of optimization solvers. This is achieved by introducing a "combinatorial loss" in addition to the regular generative loss. The combinatorial loss enforces hard constraints or add penalties for constraint violation. GenCO's involves generating a problem representation $c$. The combinatorial loss $C$ uses a MILP solver to project $c$ onto the feasible set (for hard constraints) or adds a penalty (for soft constraints). For hard constraints, the combinatorial loss is implemented via a projection that finds the closest feasible solution to $c$ allowing training of the generator without explicit constraints as the projection layer handles feasibility. For soft constraints, the combinatorial loss instead adds a penalty term based on constraint violation. The overall loss function balances the generative loss and combinatorial loss with a hyperparameter $\gamma$. For backpropagation through the combinatorial solver, differentiable MILP solvers are used. Empirical evaluations are presented on game level generation (with hard constraints) and map generation (with soft constraints on path planning efficiency). GenCO consistently generates more diverse and higher quality solutions than baselines.

**Strengths:**

* Ability to generate diverse, high-quality solutions satisfying combinatorial constraints is very useful for many design problems. This is a major strength of GenCO.

* As far as I can tell, this combination of deep generative models with combinatorial solvers is novel. It draws on the strengths of both approaches to solve a useful problem.

* Experiments demonstrate clear improvements over baselines in diversity and quality of constrained generated objects. Though it should be noted that the baselines are not too strong.

* The approach seems fairly general and can potentially be applied to many combinatorial generative tasks beyond the demonstrated applications, for instance molecular generation as discussed in the introduction.

**Weaknesses:**

* There is limited analysis of how the approach scales with problem size and constraint complexity. More extensive experiments on large and complex problems would be useful.

* The tradeoff between generative loss and combinatorial loss is not carefully analyzed. There is not enough details on how the parameter $\gamma$ is set and the impact it has on training.

* While backpropagation through combinatorial solvers is possible and works well in a lot of settings, it can be computationally expensive. Alternate approaches to incorporate solvers might be important for scalability.

* Theoretical analysis of GenCO's properties is limited. For instance, more analysis of convergence guarantees, sample complexity could strengthen the approach.

* It is not very clear to me how the method would generalize to general logical constraints, which can be important in the design scenarios considered.

* Reproducibility: The authors do not include code to reproduce the results and experimental details are discussed but details about hyperparameter selection are missing.

**Questions:**

* Could you elaborate a bit on the scalability of the approach to complex constraints and problems?

* Could you comment on how more general constraints could be incorporated in this framework?

---

> ### Author Response · Authors · 2023-11-18
>
> > There is limited analysis of how the approach scales with problem size and constraint complexity. More extensive experiments on large and complex problems would be useful.
>
> It is indeed important to add more validation. To support our claims, we included one additional benchmark: inverse photonic design, which is a real-world, practical, and large-scale problem. Our results and discussion can be found in general response (2). For now, we have included it in the appendix, and we will include it in the main text of our camera-ready revision.
>
>
> > The tradeoff between generative loss and combinatorial loss is not carefully analyzed. There is not enough details on how the parameter is set and the impact it has on training.
>
> Please review the additional results presented below. As anticipated, the SP loss decreases with an increase in gamma. However, the image quality concerning other metrics, such as density/coverage, starts to deteriorate rapidly. Eventually, at a certain gamma value, our approach generates blank or empty images, as such images result in the shortest path with zero cost. This intriguing behavior could be attributed to the combinatorial nature of the problem.
> We use cross-validation to set the gamma (and other hyperparameters). We will make this clear in the next revision of the paper.
> | Gamma        | SP Loss | Density | Coverage |
> |--------------|---------|---------|----------|
> | gamma = 0    | 36.45   | 0.81    | 0.98     |
> | gamma = 1e-4 | 27.66   | 0.93    | 0.93     |
> | gamma = 1e-3 | 23.99   | 0.94    | 0.93     |
> | gamma = 3e-3 | 23.76   | 0.75    | 0.86     |
> | gamma = 5e-3 | 18.02   | 0.49    | 0.61     |
> | gamma = 1e-2 | 0.00    | 0.00    | 0.00     |
>
> Note: gamma = 1e-2 generates blank/empty images
>
> > While backpropagation through combinatorial solvers is possible and works well in a lot of settings, it can be computationally expensive. Alternate approaches to incorporate solvers might be important for scalability.
>
> There are indeed alternative methods that can be used depending on problem type. We've employed (Sahoo et al, 2022) for Warcraft experiments, which is much faster computation-wise (and arguably more stable). Additionally, "solver-free" methods could be involved (Shah et al., 2022; Zharmagambetov et al., 2023), which will be important to use as a replacement for the differentiable optimizer as the optimization problem at hand becomes less computationally tractable.
>
>
> > Theoretical analysis of GenCO's properties is limited. For instance, more analysis of convergence guarantees, sample complexity could strengthen the approach.
>
> We only have guarantees on the feasibility of the generated solutions, it is unclear how to obtain meaningful convergence guarantees or sample complexity without resorting to unreasonable and simple assumptions.
>
>
> > It is not very clear to me how the method would generalize to general logical constraints, which can be important in the design scenarios considered.
>
> General logical constraints are possible as long as they can be expressed as the feasible region to an integer linear program which has been flexibly used to encode constraints in scheduling, routing, resource allocation, matching, time windowed delivery, coloring, packing, first-order logic, and more.
>
> > Reproducibility: The authors do not include code to reproduce the results and experimental details are discussed but details about hyperparameter selection are missing.
>
> We have clarified training details in the Supplementary files and in the Experiments section. We will add more details in the next revision. Furthermore, we are committed to enhancing openness by open-sourcing our implementation before the camera-ready, enabling our methodology to be extended to relevant domains.
>
> > Could you elaborate a bit on the scalability of the approach to complex constraints and problems?
>
> It is applicable as long as the constraints can be optimized over. That is, as long as the underlying optimization problem can be solved either exactly or with reasonable approximation. Modern solvers such as Gurobi often solve industrial-scale problems in seconds or minutes. Furthermore, there is potential to extend recent advances in "solver-free" methods (Shah et al., 2022; Zharmagambetov et al., 2023) in cases where the optimization model is expensive to solve.
>
> > Could you comment on how more general constraints could be incorporated in this framework?
>
> Our approach can handle any kind of constraints that can be expressed using optimization modeling tools that have been made differentiable, such as maxSAT, mixed integer linear programs, cone programs, quadratic programs, and many more. We refer to Kotary, James, et al. "End-to-End Constrained Optimization Learning: A Survey." 2021 for an overview of the different methods here, most of which could be readily integrated into our pipeline depending on the task at hand.

---

> > ### Comment · Reviewer_X2M2 · 2023-11-20
> > **Response to Rebuttal**
> >
> > Thanks for the clear response and I appreciate the authors adding an additional task in such a short time-frame. It is a bit difficult to gauge the complexity of the added task and judge the applicability to other large domains but the results are certainly promising. The ablations are also helpful and would be a good addition to the paper. My concerns regarding the reliance on having a good solver and being able to define constraints to the solver remain, but I have raised my score to recommend acceptance (5-> 6).

---

> > > ### Author Response · Authors · 2023-11-21
> > >
> > > We thank the reviewer for their comments and discussion, which have really pushed us to improve our work by adding an extra domain and further exploring our training dynamics.
> > >
> > > We would also like to note that in the game level generation and the inverse photonic setting, GenCO is treating the combinatorial solver as a blackbox optimizer. Indeed, in the inverse photonics setting, our solver is the blackbox brush-based solver proposed in previous work (the cited Schubert et al 2022). Due to GenCO’s flexibility, we took the domain-specific solver off the shelf to generate many unique and provably feasible photonic devices. The solver itself is intended to identify a combinatorially feasible solution that maximizes cosine similarity to a given continuous solution (with cosine similarity being a simple dot product $c^Tx$). This is to say that GenCO can still handle cases where we have access to a blackbox solver with little indication of the mathematical formulation for the underlying constraints. We simply need the solver to find a solution satisfying the combinatorial constraints and optimizing a given input objective. We will clarify our flexibility and requirements in our paper and also explain that GenCO is able to leverage various previously proposed methods for differentiating through solvers, including "solver-free" (Shah et al., 2022; Zharmagambetov et al., 2023) methods in cases where the solver is prohibitively expensive.

---

> > > > ### Comment · Reviewer_X2M2 · 2023-11-22
> > > >
> > > > Thanks for the additional clarification!

---

### Official Review · Reviewer_5GZK · 2023-11-01

**Soundness:** 3 good
**Presentation:** 3 good
**Contribution:** 3 good
**Rating:** 6
**Confidence:** 4

**Summary:**

This paper focuses on the generative tasks in which both the diversity and conformity of constraints are crucial. The objective is to make the output of the generator follow discrete/combinatorial constraints and penalize any deviation. The proposed framework enables end-to-end training of deep generative models integrated with embedded combinatorial solvers, aiming to guarantee the combinatorial feasibility of the generation while also maintaining high fidelity. The effectiveness of the proposed method is verified in generative tasks characterized by combinatorial intricacies, including game level generation and map creation for path planning, showing its superiority over previous peer methods.

**Strengths:**

1.$\ $The setup and task design studied in this article are intriguing and inspiring. In real-world scenarios, natural data often exhibit some discrete properties, which is an aspect overlooked in the realm of pure generative model research.

2.$\ $The paper is well organized and presented. The motivations and current challenges on top of the current techniques are clearly stated.

3.$\ $The methodology design is simple yet efficient. The empirical results are promising.

**Weaknesses:**

1.$\ $My primary concern regarding this paper is the selection of quantitative evaluation metrics. As a solution for generative tasks, the assessment of diversity and generation quality should draw from some classic evaluation metrics in the traditional generative model field, such as FID [1] and density/coverage [2]. While the evaluation method using a discriminator indeed holds some value, it is not an authoritative network, and this evaluation metric is single-dimensional in terms of discriminability, making its reliability less robust.

2.$\ $Some training details can be more specific. For instance, one aspect to consider is the stability and efficiency of the GAN training process. GANs are renowned for their challenging training dynamics, and the utilization of approximated gradients obtained through black-box optimization methods could potentially exacerbate the risk of training instability. Moreover, since the training data is limited, effectively training a GAN becomes a non-trivial endeavor.

[1] Gans trained by a two time-scale update rule converge to a local Nash equilibrium. NeurIPS 2017.

[2] Reliable fidelity and diversity metrics for generative models. ICML 2020.

**Questions:**

1.$\ $What about the quantitative evaluation results of traditional metrics in the generative model field such as FID and density/coverage?

2.$\ $"..., given that the levels are trained on only 50 examples, ...": How is the training dynamic of GAN with so limited data? Are there any additional efforts to stabilize training?

3.$\ $In the discussion of uniqueness in Sec. 4.1.2, it makes the adversary's task easier as it only needs to distinguish between valid discrete levels rather than continuous and unconstrained levels. But if the discriminator is too strong, it can easily lead to the phenomenon of gradient vanishing, does this make training even more challenging and unstable?

4.$\ $In fact, many generative tasks naturally exhibit certain discrete characteristics, but due to the strength of generative models, they can inherently learn these features. For example, generative models can learn that dogs have four legs. Is it possible that with a sufficiently powerful generative model, it can automatically recognize and learn these discrete features?

5.$\ $Typo: Sec. 4.2: descrbied -> described. Sec. 4.2.1: Delete "We then average the output to obtain the final loss for generator."

**Details Of Ethics Concerns:**

No ethics concerns.

---

> ### Author Response · Authors · 2023-11-18
>
> > Evaluation metrics
>
> Thanks for pointing this out. Please see our general response (3) above, where we provide additional quantitative metrics (density/coverage). We will definitely include them in our paper revision.
>
>
> > Some training details can be more specific. For instance, one aspect to consider is the stability and efficiency of the GAN training process. GANs are renowned for their challenging training dynamics, and the utilization of approximated gradients obtained through black-box optimization methods could potentially exacerbate the risk of training instability. Moreover, since the training data is limited, effectively training a GAN becomes a non-trivial endeavor.
>
> We have implemented suggested regularization techniques from (Pogancic et al. (2020); Sahoo et al. (2022)) to stabilize the training of differentiable combinatorial solvers. Also, we have clarified training details in the Supplementary files and in the Experiments section. We will add more details in the next revision. Subjectively, we observe that training our pipeline has similar complexity as training regular GANs in terms of stability, hyperparameter tuning, etc.
>
> Furthermore, we are committed to enhancing openness by open-sourcing our implementation soon.
>
> Understanding the empirical nature of the concerns raised, we have conducted additional empirical evaluations (photonic design in general response (2)) to strengthen the robustness of our claims. Additionally, we've explored the trade-off of the gamma parameter in Section 3.3 (see our response to reviewer @X2M2).
>
>
> > "..., given that the levels are trained on only 50 examples, ...": How is the training dynamic of GAN with so limited data? Are there any additional efforts to stabilize training?
>
> In game-level design, the discrete nature of GenCO’s output is beneficial in our scenario. Specifically, there are finite discrete objects that the GAN can generate, and thus it is easier to avoid the issues of mode collapse compared to a traditional GAN, which can effectively hide solutions in the continuous space. Indeed, this is shown in practice as many of the continuous solutions map to the same discrete solution after postprocessing.
>
>
> > In the discussion of uniqueness in Sec. 4.1.2, it makes the adversary's task easier as it only needs to distinguish between valid discrete levels rather than continuous and unconstrained levels. But if the discriminator is too strong, it can easily lead to the phenomenon of gradient vanishing, does this make training even more challenging and unstable?
>
> As in the previous level generation paper, we use Wasserstein GAN (WGAN), which was designed to prevent vanishing gradients as there is always some incoming gradient imposed on the generated solution as long as the generated solution is different from the known solutions and the adversary hasn’t become degenerately poor. Overall, we only have access to 50 training levels, and GenCO is able to perform well in this low data regime by still generating many unique solutions while the postprocessing approach fails to do so.
>
>
> > In fact, many generative tasks naturally exhibit certain discrete characteristics, but due to the strength of generative models, they can inherently learn these features. For example, generative models can learn that dogs have four legs. Is it possible that with a sufficiently powerful generative model, it can automatically recognize and learn these discrete features?
>
> While powerful generative models can capture certain discrete features, there are some limitations to the models, even with large capacity. For example, at the time of writing, large models like stable diffusion or DALL E 2 suffer from generating images of people with incorrect numbers of fingers or toes, or where their pose doesn't match that of their reflection in a mirror unless specifically fine-tuned and instructed for such tasks. Furthermore, going beyond image/video generation, for other design tasks with mathematically well-defined constraints (e.g., industrial design), GenCO can make full use of the domain-specific constraints. It can flexibly and explicitly encode such constraints and eliminate the possibility of generating infeasible solutions, regardless of the size of the datasets. This could lead to better performance with the same computational costs and model capacity.

---

> > ### Comment · Reviewer_5GZK · 2023-11-23
> > **Thanks for your response**
> >
> > Thanks for the reply. It is still counter-intuitive to me that WGAN can be well trained by 50 samples without any technique tailored for few-shot scenarios. Are you planning to release the code at some point?

---

> > > ### Author Response · Authors · 2023-11-23
> > >
> > > Thank you, we will release the code to reproduce the results. We don't use any special techniques for training WGAN in the low-data setting that weren't introduced in previous work. Indeed, we use the same training framework and architecture presented in previous work (Zhang et al 2020) to ensure that we are comparing against a method tuned for this setting. The code for training their approach is available publicly here: https://github.com/icaros-usc/milp_constrained_gan/blob/master/launchers/zelda_gan_training.py. The main small tweak they use that may improve low data performance is discriminator parameter clamping to be between -0.01 and 0.01. Otherwise, they simply train a standard using WGAN using rmsprop.
> > >
> > > We would note that the previous work was also trained using the same dataset of 50 human-authored levels. We conjecture that our performance improvement is mainly possible because the generator doesn't need to dedicate capacity to learning to satisfy constraints to better match the data, and because the discriminator doesn't need to dedicate capacity to determine feasibility.
> > >
> > > We will clarify the tweaks inherited from previous work in our experimental details to make the overall pipeline clearer and provide further intuition for low data performance.

---

> > > > ### Comment · Reviewer_5GZK · 2023-11-23
> > > > **Thanks**
> > > >
> > > > Thanks for the clarification. I would like to uphold my current rating.

---

### Official Review · Reviewer_qozd · 2023-11-02

**Soundness:** 2 fair
**Presentation:** 2 fair
**Contribution:** 2 fair
**Rating:** 5
**Confidence:** 3

**Summary:**

This paper proposes GenCO that generates diverse solutions, instead of the best solution given by traditional solvers, for design poblems with combanitorial nature. Specifically, it deals with either hard or soft constraints by introducing  a combanitorial loss.

**Strengths:**

1. The paper is well-motivated.
2. The results are good according to the case study, while more metrics are expected for better evaluation.

**Weaknesses:**

1. The so-called combinatorial loss is unsurprising, as it has been used in many areas, though without a uniform name or formulation. For example, in areas such as molecule design and chip design, it is usual to use the panelty in reward design for hard or soft constraints. It is also used in solving combinatorial problems to guide the feasible and quality of solutions. This kind of methods are like the Lagrangian multiplier method, and the idea of modeling the constraints as a loss term is straightforward. There are also many methods targeting on solving combinatorial problems, such as mixed-integer programmings, by generative models, where the combinatorial constraints are also considered. This paper is more like a gatherer of those methods. If I am wrong, the authors may want to further emphasise the technical contribution.
2. In experiments, the authors report the loss values for evaluation. However, the loss function may not reflect the generation quality precisely, and it is more like a surrogate metric instead of the final goal. Some other metrices for evaluation should be introduced.
3. Baselines are not strong enough. When considering the specific scenarios such as game design, it is expected to compare the proposed method with the SOTA method tailored for this task, instead of only considering the GAN+MILP baseline, to demonstrate the effectiveness of GenCO.
4. If the authors claim that they propose a framework for design problems, it is expected to conduct more experiments such as molecule or chip design. Current experiments are not convincing enough.

**Questions:**

See weaknesssed.

---

> ### Author Response · Authors · 2023-11-18
>
> > Combinatorial loss has been used in many areas, e.g. molecule and chip design, where it is usual to use the penalty in reward design for hard or soft constraints
>
> While it's true that penalizing constraint violation has been employed in various domains like molecule and chip design, it is critical to emphasize that they typically do not guarantee feasibility. One possible reason for this is that in molecule or chip design settings, the constraints are often provided as blackbox simulators that are difficult to represent using constraint optimization solvers, thus, it is difficult to guarantee that solutions satisfy constraints such as synthesis accessibility or water lipid solubility.
>
> Our work stands out as we uniquely combine generative models (GANs/VAEs) and combinatorial solvers with feasibility guarantees and propose an end-to-end training algorithm for the entire pipeline.
>
> Also, please refer to our general response regarding the novelty and importance of our work, as well as our experiments on the photonic device design setting to demonstrate generalizable performance.
>
> > ​​This kind of methods are like the Lagrangian multiplier method, and the idea of modeling the constraints as a loss term is straightforward.
>
> Thank you for your observation. While the concept of using combinatorial optimization as a loss term may seem reminiscent of the Lagrangian multiplier method, it's important to emphasize a key distinction in our approach. Unlike the Lagrangian multiplier method, which lacks feasibility guarantees, our method provides explicit feasibility guarantees throughout training. Furthermore, in discrete optimization cases, handling Lagrangian functions and multipliers can be non-trivial, and there may not be feasibility guarantees in cases where the continuous relaxation is very loose. In contrast, our approach incorporates hard constraints seamlessly using a differentiable combinatorial layer in the pipeline, allowing for end-to-end training. These key differences set our methodology apart in terms of both feasibility assurances and empirically improved performance.
>
>
> > There are also many methods targeting on solving combinatorial problems, such as mixed-integer programmings, by generative models, where the combinatorial constraints are also considered.
>
> These methods are mainly concerned with solving a single Mixed Integer Program rather than generating multiple solutions that satisfy the combinatorial constraints. Our goal is to satisfy all three conditions on solutions: diversity, solution quality, and feasibility. Furthermore, these problems generally concern solving a single well-defined problem where all objective coefficients and constraints are known. In our settings, the problem is underspecified in that for the game-level design, inverse photonic design, or shortest path map settings, we don’t have a fixed problem formulation that we are trying to solve.
>
> > This paper is more like a gatherer of those methods. If I am wrong, the authors may want to further emphasise the technical contribution.
>
> Hopefully, our answers above and general response regarding our technical contribution clarified your concerns. Furthermore, as we demonstrate in the game level design and inverse photonic design setting, GenCO is the only approach that is capable of providing many unique solutions which are all guaranteed to satisfy combinatorial constraints, whereas previous approaches that treat optimization as a postprocessing step yield degenerate solutions.
> > Other evaluation metrics for measuring the generation quality
> Thanks for pointing this out. Please see our general response (3) above where we provide additional quantitative metrics (density/coverage). We will definitely include them in our paper revision.
>
> > Baselines are not strong enough.
>
> From our review of the literature, we found that the GAN+MILP was the only SOTA baseline that was also able to guarantee feasibility of the generated solutions in the game-level design setting. We furthermore have pure GAN results that demonstrate that these approaches don't guarantee feasibility. The same goes for the penalty version of GenCO.
>
> > If the authors claim that they propose a framework for design problems, it is expected to conduct more experiments such as molecule or chip design. Current experiments are not convincing enough.
>
> It is indeed important to add more validation. To support our claims, we included one additional benchmark: inverse photonic design, which is a real-world, practical, and large-scale problem. Our results and discussion can be found in general response (2). For now, we have included it in the appendix, and we will include it in the main text of our camera-ready revision.

---

> > ### Comment · Reviewer_qozd · 2023-11-22
> >
> > Thanks for the authors' responses, which help me understand the contributions better. However, I still have some questions.
> > 1. In my understanding, the authors propose a framework that first generates a problem description, and leverages a solver to find a problem solution that is "clear" to the description in terms of the dot product similarity. It seems that $h(c)$ is just like a postprocess (or in other word, the process of obtaining the generated data from a latent vector), but GenCO conducts the postprocess during training, instead of after training. In this way, the trained generator is aware of what we obtain after postprocessing. Am I right? If so, the calling of a solver is just a special case.
> > 1. This paper considers generating a vector "x", so it can be ontained by minimizing the dot conduct. However, what if the generated data has a complex form that cannot be easily represented as a vector "x"? How to define $h(c)$ then? For example, can you briefly describe how to define $h(c)$ if I want to use GenCO to generate graphs?
> > 1. Is it always easy to solve the MILP $\arg\min_{x\in\Omega}c^{\top} x$? As in many scenarios, the feasible domain may be complex, and may not be easily represented as a MILP.
> > 1. I still think the baselines are not strong enough, so that I can hardly understand how powerful is the trained model. Is it possible to conduct experiments on more common tasks where more SOTA approaches---maybe carefully designed for those tasks with or maybe with written rules---can be compared with?
> >
> > BTW, 'z' in Algorithm 3 Line 10 should be 'x'?

---

> > > ### Author Response · Authors · 2023-11-23
> > >
> > > We thank the reviewer for their additional comments and feedback. We will use these questions to clarify our work and applicability. To your questions:
> > >
> > > > 1. In my understanding, the authors propose a framework that first generates a problem description, and leverages a solver to find a problem solution that is "clear" to the description in terms of the dot product similarity. It seems that is just like a postprocess (or in other word, the process of obtaining the generated data from a latent vector), but GenCO conducts the postprocess during training, instead of after training. In this way, the trained generator is aware of what we obtain after postprocessing. Am I right? If so, the calling of a solver is just a special case.
> > >
> > > In our approach, we perform differentiable processing of the generative network's output during training to enable end-to-end training. Gradients from the generative loss (discriminator for GAN or reconstructive loss for VAE) are backpropagated through the solver, aligning the generator with the downstream solver and ensuring compliance with specified combinatorial constraints throughout training. We demonstrate that incorporating the solver as a differentiable layer in the training process empirically improves generative performance by preventing mode collapse (generating unique solutions), generating more realistic solutions (as evaluated by adversarial discriminator), and can be flexibly incorporated even when we only have access to a blackbox solver (as in inverse photonic design).
> > >
> > > Note that "calling a solver" is an important strategy that completely changes the behaviors of the generators, which we regard as the key contribution that solves the constrained generation problems well. We want to emphasize that many successful deep learning applications basically leverage a function approximator, compared to basic approximators like logistic regression, and can be regarded as a “special case”. So this should not be counted as a negative point towards our contribution.
> > >
> > >
> > > > 2. This paper considers generating a vector "x", so it can be ontained by minimizing the dot conduct. However, what if the generated data has a complex form that cannot be easily represented as a vector "x"? How to define then? For example, can you briefly describe how to define if I want to use GenCO to generate graphs?
> > >
> > > Note that a vector “x” can represent a rich set of objects, through different encodings. E.g., a vector can represent a word (“word embedding”) (Church 2017), an image (“image embedding”) (Krizhevsky et al. 2012), a sentence (“sentence embedding”) (Joulin 2016), a graph (“graph embedding”), a code snippet (Alon 2019), etc. The notation of “embedding” in a latent high-dimensional space, and how any object can be encoded into such embedding vectors, is actually a key concept in the deep learning era.
> > >
> > > For instance, to represent a graph with a vector x, we could take an off-the-shelf graph generation backbone, such as GraphVAE (Simonovsky & Komodakis 2018) which can encode a true graph to an embedding vector in the latent space, and decode any such vector into a probabilistic graph. From the decoded graph, we get edge probabilities and use a solver to find the most likely graph (subset of edges) that satisfies certain combinatorial constraints. This decoded and feasible graph would then be used to compute a generative
> > > (reconstruction or adversarial) loss against the real graph from data. We then backpropagate through the solver to update the generative model.
> > >
> > > - Church, Kenneth Ward. "Word2Vec." Natural Language Engineering 23.1 (2017): 155-162.
> > > - Krizhevsky, Alex, Ilya Sutskever, and Geoffrey E. Hinton. "Imagenet classification with deep convolutional neural networks." Advances in neural information processing systems 25 (2012).
> > > - Joulin et al. Bag of Tricks for Efficient Text Classification. 2016.
> > > - Uri Alon, Meital Zilberstein, Omer Levy, and Eran Yahav. 2019. Code2vec: learning distributed representations of code. Proc. ACM Program. Lang. 3, POPL, Article 40 (January 2019), 29 pages. https://doi.org/10.1145/3290353
> > > - Simonovsky, Martin, and Nikos Komodakis. "Graphvae: Towards generation of small graphs using variational autoencoders." Artificial Neural Networks and Machine Learning–ICANN 2018: 27th International Conference on Artificial Neural Networks, Rhodes, Greece, October 4-7, 2018, Proceedings, Part I 27. Springer International Publishing, 2018.

---

> > > > ### Author Response · Authors · 2023-11-23
> > > >
> > > > > 3. Is it always easy to solve the MILP? As in many scenarios, the feasible domain may be complex, and may not be easily represented as a MILP.
> > > >
> > > > While MILP is generally NP-Hard, solvers such as Gurobi are known to handle industry-level problems in domains ranging from “energy, finance, health, manufacturing, military, transportation, and in almost any imaginable domain where decisions are made” and “most Fortune 500 companies use integer programming in some aspects of their business” (Nemhauser 2013). Additionally, our method isn’t entirely dependent on using a MILP solver as evidenced by the inverse photonics domain which uses a domain-specific blackbox solver to find verifiably feasible solutions. This is done because the domain-specific solver leverages domain knowledge that improves solve speed and is readily available.
> > > >
> > > > Nemhauser, George L. “Integer programming: The global impact.” (2013).
> > > >
> > > > > 4. I still think the baselines are not strong enough, so that I can hardly understand how powerful is the trained model. Is it possible to conduct experiments on more common tasks where more SOTA approaches---maybe carefully designed for those tasks with or maybe with written rules---can be compared with?
> > > >
> > > >
> > > > We would be happy to evaluate GenCO against more baselines on more tasks. However, given that pure ML approaches cannot guarantee satisfaction of general combinatorial constraints, the area of generating objects satisfying arbitrary combinatorial constraints has not seen much work or has not yielded ML methodologies that are likely to generate feasible solutions. The only work we could find approaching this problem of generating combinatorially constrained objects is the referenced game level design approach (Zhang et al. 2020).
> > > >
> > > >
> > > > Our approach also has potential application in many constrained design settings similar to inverse photonic design (which we have now), such as designing circuits (Guo et al. 2019). To this end, it appears that current SOTA approaches being pure ML are not even able to generate reliably feasible solutions, with standard generative models generating at most 40% of the solutions being feasible. Given that the code and data aren’t publicly available, we weren’t able to run experiments in this domain. However, this seems like a promising additional domain.
> > > >
> > > >
> > > > Additionally, we apply a relaxation technique mentioned by several reviewers in Warcraft map generation (see “GAN+costNN” in appendix D), which shows pretty comparable performance in terms of density/coverage.
> > > >
> > > >
> > > > Zhang, Hejia, et al. "Video game level repair via mixed integer linear programming." Proceedings of the AAAI Conference on Artificial Intelligence and Interactive Digital Entertainment. Vol. 16. No. 1. 2020.
> > > >
> > > > Guo, Tinghao, Daniel Herber, and James T. Allison. "Circuit synthesis using generative adversarial networks (Gans)." AIAA Scitech 2019 Forum. 2019.
> > > >
> > > >
> > > > >BTW, 'z' in Algorithm 3 Line 10 should be 'x'?
> > > >
> > > > Thank you for finding this, we will have it fixed

---

> > > > > ### Comment · Reviewer_qozd · 2023-11-23
> > > > >
> > > > > Thanks for the further responses. Sorry but I still cannot understand something.
> > > > > 1. Regarding the generation of a vector "x". Do you mean that GenCO may rely on a kind of pretrained encoder-decoder like a VAE model, for obtaining the embedding vector and decodiing the vector into the objective to be generated? And GenCO is like a model that searches in the latent space? If so, how to express the constraints in the objective space? In other word, if the constraints are complex, and with an off-the-shelf encoder, how could we properly find the domain $\Omega$ in the latent space?
> > > > > 1. Regarding solving the MILP. In real-world applications the constraints may be complex, so that the domain $\Omega$ could not be easily modeled as a polygon, so the problem may not even be a MILP. How to tackle this?
> > > > > 1. I understand that previous pure ML approaches cannot work well on this job. What about those rule-based or heuristic methods? It would help readers better understand the difficulty of the tasks.
> > > > >
> > > > > I would like to raise my score if the authors can explain these questions clearly.

---

> > > > > > ### Author Response · Authors · 2023-11-23
> > > > > >
> > > > > > We thank the reviewer for their effort in providing continued feedback. To your specific questions:
> > > > > >
> > > > > > > 1. Regarding the generation of a vector "x". Do you mean that GenCO may rely on a kind of pretrained encoder-decoder like a VAE model, for obtaining the embedding vector and decodiing the vector into the objective to be generated? And GenCO is like a model that searches in the latent space? If so, how to express the constraints in the objective space? In other word, if the constraints are complex, and with an off-the-shelf encoder, how could we properly find the domain in the latent space?
> > > > > >
> > > > > > We don’t search in the latent space (space directly after the VAE encoder), instead, we search over the design space (taking the values directly after the decoder to use for objective coefficients and finding a solution satisfying the constraints).
> > > > > >
> > > > > > Overall, GenCO takes the values that the generative backbone would use to approximate an example from the data distribution (output of decoder for VAE and output of generator for GAN) and finds a feasible solution in a differentiable manner that is close to the neural output and which satisfies the hard constraints. Here, all optimization is done in the “design space” such as the space of binary decisions of whether a grid cell should be a specific tile type in game-level design, or whether a grid cell should be filled in the photonics design setting. In game-level design, the solver takes as input log probabilities that each cell is of a given tile type and then finds the most likely game level satisfying playability constraints. In the inverse photonic design, the solver takes a continuous matrix of arbitrary height and width and outputs a binary matrix that satisfies the manufacturability constraints. In the constrained graph generation setting from our earlier comment, the design space would be the space of graphs defined by binary decisions on each edge determining whether an edge should exist or not.
> > > > > >
> > > > > > Note that the design space can change dimensionality depending on the input, so if the GAN generated coefficients with a specific width and height, the solver would run with that specific width and height to ‘correct’ the input image. Similarly, if the graph generator creates a probabilistic adjacency matrix with a specific number of nodes, then the solver would run with that specific number of nodes, adjusting as needed if the generator generates graphs of different sizes. We will clarify this in our paper to make the design space and objective coefficient space clearer.
> > > > > >
> > > > > > > 2. Regarding solving the MILP. In real-world applications the constraints may be complex, so that the domain could not be easily modeled as a polygon, so the problem may not even be a MILP. How to tackle this?
> > > > > >
> > > > > > While we acknowledge that many real-world applications may have complex (nonlinear or blackbox) constraints, many other real-world applications contain linear constraints on real/integer variables and are important problems to solve (e.g., scheduling, planning, designing). The MILP solver does not aim to solve every real-world problem but focuses on a subset of them.
> > > > > >
> > > > > > Furthermore, the GenCO framework can leverage combinatorial solvers beyond MILP. For example, the inverse photonic design setting doesn’t use a MILP solver and instead uses a domain-specific brush-based blackbox solver to find a solution satisfying the constraints. As long as we have access to some (potentially blackbox) solver that can take objective coefficients and produce a verifiably feasible solution, GenCO can leverage it to produce diverse and feasible solutions that are aligned with the data distribution. The exact requirements of the differentiable blackbox technique that we use in some of our domains (Pogančić et al 2019) are that the solver needs to find a solution optimizing a linear objective over a discrete feasible region. In this case, the feasible region may not be a polygon, but GenCO will still be able to leverage any applicable solver. Very recent "solver-free" differentiation methods (Shah et al., 2022; Zharmagambetov et al., 2023) enabled differentiation for arbitrary optimization solvers, which GenCO can leverage depending on the problem at hand.
> > > > > >
> > > > > >
> > > > > > - Pogančić, Marin Vlastelica, et al. "Differentiation of blackbox combinatorial solvers." International Conference on Learning Representations. 2019.
> > > > > > - Shah, Sanket, et al. "Decision-focused learning without decision-making: Learning locally optimized decision losses." Advances in Neural Information Processing Systems 35 (2022): 1320-1332.
> > > > > > - Zharmagambetov, Arman, et al. "Landscape Surrogate: Learning Decision Losses for Mathematical Optimization Under Partial Information." Thirty-seventh Conference on Neural Information Processing Systems. 2023.

---

> > > > > > > ### Author Response · Authors · 2023-11-23
> > > > > > >
> > > > > > > > 3. I understand that previous pure ML approaches cannot work well on this job. What about those rule-based or heuristic methods? It would help readers better understand the difficulty of the tasks.
> > > > > > >
> > > > > > > We will include a discussion of rule-based or heuristic methods for generating combinatorially constrained objects. Generally, the limitation of these methods is that they are unable to leverage data to match a data distribution and are geared more towards very uniformly generating objects that satisfy combinatorial constraints, such as in constrained graph generation (Shine & Kempe 2019), game-level design (Sorenson et al. 2011), or topology optimization (Bendsøe & Kikuchi 1988). These approaches generally require the practitioner to set parameters of the generative model with arduous tuning of the model to promote specific patterns. On the other hand, GenCO is able to train generative neural networks from data to automatically learn patterns that make up realistic examples. Additionally, in certain cases, GenCO can learn coefficients for these solvers, which often generate solutions individually, resulting in solutions that match a given data distribution and still have the theoretical guarantees proven by these rule-based methods.
> > > > > > >
> > > > > > > - Shine, Alana, and David Kempe. "Generative Graph Models based on Laplacian Spectra?." The World Wide Web Conference. 2019.
> > > > > > > - N. Sorenson, P. Pasquier and S. DiPaola, "A Generic Approach to Challenge Modeling for the Procedural Creation of Video Game Levels," in IEEE Transactions on Computational Intelligence and AI in Games, vol. 3, no. 3, pp. 229-244, Sept. 2011, doi: 10.1109/TCIAIG.2011.2161310.
> > > > > > > - Bendsøe, Martin Philip, and Noboru Kikuchi. "Generating optimal topologies in structural design using a homogenization method." Computer methods in applied mechanics and engineering 71.2 (1988): 197-224.

---

### Author Response · Authors · 2023-11-18
**General Response 1**

## General Response
We thank the reviewers for their constructive feedback, which we have used to improve our work. We hope that if we adequately address your comments, you will consider increasing your score. We respond to the general comments below:

## (1) Novelty and motivation
GenCO is the first approach that can generate many unique solutions that strictly conform to flexible and nontrivial combinatorial constraints through end-to-end (e2e) training with existing combinatorial solvers (e.g., mixed integer linear programming (MILP) solvers like Gurobi). It tightly integrates the two seemingly unrelated fields, neural generative techniques like GAN and VQVAE with flexible combinatorial solvers, which is novel to our best knowledge.

More specifically,
GenCO generates solutions that strictly satisfy the combinatorial constraints and can be trained in an e2e differentiable manner. In comparison, prior methods either relax the combinatorial constraints during training, which leads to infeasible solutions, or simply postprocess generated examples, which misaligns the training and deployment process by sacrificing the end2end property. We show that this has profound implications for empirical performance.
GenCO can still be configured to enforce soft constraints besides enforcing strictly feasible solutions.
GenCO leverages the flexible language of mathematical programming, specifically mixed integer linear programming (MILP) and linear programming (LP), and gives the practitioner great feasibility under the same mathematical framework. In comparison, prior works need to craft problem-specific loss functions to promote feasible solutions, or create hand-crafted model architectures for each individual setting.

In summary, GenCO introduces a new way of thinking about generating objects with combinatorial properties by fundamentally altering how generative models handle constraints. The guarantee of feasibility throughout training, coupled with the unification of constraint and penalty formulations, positions GenCO as an exciting and transformative approach with far-reaching implications for the field of generative modeling.

---

> ### Author Response · Authors · 2023-11-18
> **(2) additional domain**
>
> ## (2) Additional domain: Diverse photonic design
>
> We further demonstrate the effectiveness of GenCO for an inverse photonic design setting over a generative + postprocess fix baseline. See Appendix E.
>
> The inverse photonic design problem [1] asks how to design a device consisting of filled and void space to route wavelengths of light from an incoming location to desired output locations at high intensity. Here, the feasible region consists of deciding whether to fill or void a grid cell while satisfying manufacturing constraints that a die with a specific shape must be able to fit in every filled and void area. Specifically, the filled and void regions, respectively should be able to be represented as a union of the die shape. The objective function here consists of a nonlinear but differentiable simulation of the light using Maxwell’s equations. Previous work [1] demonstrated an approach for finding a single optimal solution to the problem. However, we propose generating a diverse collection of high-quality and unique solutions using a dataset of known solutions.
>
> Here, we instantiate GenCO using a vector quantized variational autoencoder (VQVAE) [2] generative backbone. Here, the autoencoder is fed in a known solution and then uses neural networks for the encoder and decoder. The continuous decoded object is then fed into the differentiable solver to ensure that the generated solution is feasible. This feasible solution is then used to calculate the reconstruction loss to the known solution. Furthermore, in this setting, we have a penalty term that consists of the simulation of Maxwell’s equations. As such, we ablate the generative and penalty losses of GenCO: whether or not to train using the reconstruction loss, and whether or not to penalize generated solutions based on Maxwell’s equations. We consider a baseline here of training the same generative architecture without the combinatorial optimization layer and then postprocessing generated examples during evaluation using a combinatorial solver. We demonstrate the results below, including the percentage of unique discrete solutions that are generated, the average loss evaluated using Maxwell’s equations, as well as the density and coverage with respect to the training dataset. The dataset of 100 examples is obtained by expensively running previous work [1] until it reaches an optimal loss 0 solution. We evaluate performance by generating 1000 feasible examples.
>
> | Approach            	| % Unique (higher better) | Avg Solution Loss (lower better) | Density (higher better) | Coverage (higher better) |
> |-----------------------|----------|-------------------|---------|----------|
> | VQVAE + postprocess 	| 0.306	| 1.244         	| 0.009   | 0.006	|
> | GenCO (reconstruction only)  	| 1    	| 1.155         	| 0.148   | 0.693	|
> | GenCO (objective only) | 0.466	| 0             	| 0.013   | 0.036	|
> | GenCO (reconstruction + objective)| 1   	| 0             	| 0.153   | 0.738	|
>
> These results make our claim of empirical performance more robust in that the postprocessing approach obtains very few unique solutions, which all have high loss and, furthermore, don’t cover the data distribution well. This is largely due to the method not being trained with the postprocessing in the loop, and thus, although it closely approximates the data distribution with continuous solutions, when these continuous solutions are post-processed to be made feasible, they are no longer representative of the data distribution and many continuous solutions collapse to the same discrete solution.
>
> Here GenCO (reconstruction only) gives many unique solutions that closely resemble the data distribution. However, the generated devices fail to perform optimally in the photonic task at hand. Disregarding the reconstruction loss and only training the decoder to generate high-quality solutions yields high-quality solutions that are unfortunately not diverse. Combining both the generative reconstruction penalty as well as the nonlinear objective, GenCO is able to generate a variety of unique solutions that optimally solve the inverse photonic design problem while having good density and coverage for the data distribution.
>
>
> [1] Schubert, Martin F., et al. "Inverse design of photonic devices with strict foundry fabrication constraints." ACS Photonics 9.7 (2022): 2327-2336.
>
> [2] Van Den Oord, Aaron, and Oriol Vinyals. "Neural discrete representation learning." Advances in neural information processing systems 30 (2017).

---

> ### Author Response · Authors · 2023-11-18
> **(3) additional generative metrics**
>
> ## (3) Additional generative evaluation metrics
> Several reviewers expressed concerns about the choice of quantitative evaluation metrics for diversity aside from unique percentage and realism in our study. We acknowledge that relying solely on Adversary (or Discriminator) loss may not adequately capture the generation quality. Therefore, we have included traditional metrics commonly used in the generative models field. Specifically, we have incorporated the suggested density/coverage metrics from https://arxiv.org/abs/2002.09797 across all experiments. It's important to note that these adjustments are in agreement with our overall findings: the density/coverage remains comparable to the baselines, while GenCO demonstrates 1) a higher rate of unique solutions, 2) feasibility guarantees, and 3) a significantly improved downstream task loss in the shortest path and inverse photonic settings.
>
> Note that our neural adversarial score can be advantageous over fixed metrics such as density/coverage, since it evolves with the training and can overcome some tricky generation that looks good on the metrics, but fails in practice. For example, in game level design, empty levels yield higher density/coverage scores (see tables below) than any meaningful generative model, because the exemplar game level contains many empty blocks. Using a neural adversary keeps pushing the generation to be similar to the training set with respect to high-level embeddings, and avoids such issues.
>
>
> Zelda game level design comparison (constrained generator)
>
> | Approach                | Density | Coverage | % Unique |
> |-------------------------|---------|----------|----------|
> | GAN + MILP              | 0.07  | 0.94   | 0.52     |
> | GenCO fixed adversary   | 0.05  | 0.98   | 0.22     |
> | GenCO updated adversary | 0.06  | 0.82   | 0.995    |
> | Empty                   | 2.5     | 1.0      | 0        |
>
>
> Warcraft map generation with shortest path (penalized generator)
>
> | Approach        | Density | Coverage | SP Loss |
> |-----------------|---------|----------|---------|
> | Ordinary GAN    | 0.81    | 0.98     | 36.45   |
> | GAN + cost NN   | 1.09    | 0.98     | 35.61   |
> | GenCO (ours)    | 0.94    | 0.93     | 23.99   |

---

### Author Response · Authors · 2023-11-21

Dear reviewers,

We hope our responses have resolved your concerns. We have diligently worked to respond to the points raised by the reviwers and have submitted a revision. We believe that your reviews and these changes strengthen the overall quality of the paper, which has been positively received by Reviewer X2M2. Your feedback has been invaluable in refining our work. If you find the responses and revisions align well with the paper's objectives and address your initial concerns, we are hopeful that an adjustment in the score could reflect these improvements.
Please feel free to ask if you have more questions or if there's anything else we can provide to support your evaluation.

Thank you!

---

### Meta-Review · Area_Chair_MkWf · 2023-12-14

**Metareview:**

This paper introduces a pipeline for training generative models that satisfy a predefined set of constraints. Of particular import to the method is its use of differentiable solvers to project generated quantities onto the feasible set; this is done not just as a test-time postprocessing, but rather at train-time with the entire process learned end-to-end. Compared to other deep generative model approaches to generation subject to constraints, this seems quite helpful, particularly in directing models away from continuous/relaxed candidates which would project down to the same feasible instance.

The reviewers had a large number of initial concerns, regarding choice of benchmarks, problem domains, and evaluation metrics, as well as the overall level of novelty and originality.

The author response has addressed these concerns quite well, and there are a number of changes to the draft (including more proposed), as well as an additional experiment. This is excellent, and three reviewers did raise their scores, although two of them still to recommend rejection.

All in all given the scope of the changes, despite the movement in the reviewer scores towards acceptance I would suggest that the new manuscript which takes these changes into account be submitted to the next relevant conference deadline, to be reviewed in its current form.

**Justification For Why Not Higher Score:**

The updates to the paper in the author rebuttal go a long way towards alleviating reviewer concerns, but some reviewers were nonetheless still arguing to reject, and the changes were fairly substantial (perhaps sufficient to warrant a fresh round of review).

**Justification For Why Not Lower Score:**

N/A

---

### Decision · Program_Chairs · 2024-01-16

Reject